# Characterization of Protein–Membrane Interactions in Yeast Autophagy

**DOI:** 10.3390/cells11121876

**Published:** 2022-06-09

**Authors:** Kelsie A. Leary, Michael J. Ragusa

**Affiliations:** Department of Chemistry, Dartmouth College, Hanover, NH 03755, USA; kelsie.a.leary.gr@dartmouth.edu

**Keywords:** autophagy, membrane binding proteins, yeast

## Abstract

Cells rely on autophagy to degrade cytosolic material and maintain homeostasis. During autophagy, content to be degraded is encapsulated in double membrane vesicles, termed autophagosomes, which fuse with the yeast vacuole for degradation. This conserved cellular process requires the dynamic rearrangement of membranes. As such, the process of autophagy requires many soluble proteins that bind to membranes to restructure, tether, or facilitate lipid transfer between membranes. Here, we review the methods that have been used to investigate membrane binding by the core autophagy machinery and additional accessory proteins involved in autophagy in yeast. We also review the key experiments demonstrating how each autophagy protein was shown to interact with membranes.

## 1. Introduction

Macroautophagy (hereafter autophagy) is a degradation process where eukaryotic cells break down and recycle their cytosolic components. As such, autophagy is essential for maintaining cellular homeostasis, and its dysfunction has been linked to diseases such as cancer and neurodegeneration [1]. Autophagy captures cytosolic material in double-membrane vesicles (termed autophagosomes) and delivers this cargo to the vacuole, in yeast, for degradation. In yeast, this process requires the following proteins or complexes, termed the core autophagy proteins: (1) the Atg1-initiation complex, (2) the PI3-kinase complex, (3) the transmembrane protein Atg9, (4) the Atg2-Atg18 complex, (5) the Atg12~Atg5 conjugation system, and (6) the Atg8~PE conjugation system. There are extensive reviews that focus on the molecular mechanisms of these complexes [2,3,4]. As such, we will not review the function of each autophagy complex in detail and instead provide a summary of the proteins in each complex and their roles in autophagy in Table 1.

Of note, this review will focus on the proteins required for autophagosome biogenesis. While there are additional proteins that are involved in autophagosome–vacuole fusion, these proteins will not be discussed in this review [5,6].

Autophagosomes are generated through the fusion of small precursor vesicles, which form an initial membrane sheet termed the phagophore [7,8]. The phagophore is then expanded via new lipid synthesis, lipid transport, and possibly through the fusion of more vesicles [7,8,9,10,11,12,13,14]. Despite the complex membrane biogenesis events that occur during autophagy, Atg9 has been identified as the sole transmembrane protein among the core autophagy proteins. Instead, a large fraction of autophagy proteins have been identified as cytosolic proteins that associate peripherally with membranes in vivo, where they bind to, tether, or structure membranes [15].

A large diversity of techniques is typically used to study protein–membrane interactions, and each of these techniques provides distinct information about these interactions. Consequently, there are large differences in how each of the core autophagy proteins have been shown to interact with membranes. While there are review articles focusing on the autophagy proteins that sense curvature and bind membranes [16,17], membrane binding domains in these proteins [15,18], and lipid composition that affords specificity of protein membrane interactions [19], none of these reviews focuses on the techniques utilized to study protein–membrane interactions. As such, this review article focuses on the techniques used to characterize protein–membrane interactions that have been specifically applied to study yeast autophagy proteins. We will begin this review with a summary of the different techniques that have been used for investigating these interactions, including how each technique works, advantages, and drawbacks/considerations. We will then discuss how these techniques have been applied to study each core autophagy protein and some additional autophagy factors, which is summarized in Table 2. 

## 2. In Cell Techniques to Monitor Protein–Membrane Interactions

Studying protein–membrane interactions is challenging due to the heterogeneity of the lipid composition, shape, and size of membranes in cells. This can often make it difficult to replicate some of these binding events in vitro. As such, many protein–membrane interactions are initially characterized in cells using qualitative techniques, which do not provide information about binding affinities or stoichiometries. One major benefit of all of the techniques in this section is that they do not require purified protein and can therefore be performed by detecting the endogenous or overexpressed protein of interest in cells using antibodies to the endogenous protein, tags, such as FLAG or HA, which can be detected via an antibody to these tags, or fluorescent tags, such as GFP or RFP. 

### 2.1. Subcellular Localization by Fluorescence Microscopy

Subcellular localization uses fluorescence microscopy to visualize the colocalization of fluorescently tagged proteins with fluorescently labeled membrane compartments in cells. Typically, this is conducted in the yeast *Saccharomyces cerevisiae* by chromosomally tagging proteins with a fluorophore, such as GFP, RFP, or other fluorescent protein tags. Alternatively, proteins may be expressed with fluorescent tags from centromeric plasmids, where the protein is expressed via its endogenous promoter, allowing for near endogenous protein expression levels from a plasmid. Different organelles can be stained simultaneously with the protein to visualize if there is colocalization, which may suggest a possible interaction with the membrane. Some examples of organelle markers include MitoTracker/Mito-BFP for mitochondria [26,76] FM4-64 for vacuoles [27,77], and Ser-Lys-Leu (SKL) targeting sequence for peroxisomes tagged with fluorophores such as GFP or RFP [78]. In the case of autophagy, the association of a protein with a membrane is typically suggested by the presence of a punctate structure colocalized with an organelle. Further confirmation can be achieved via the mutation of residues that are predicted to be membrane-binding. The release of the protein into the cytosol upon mutation of these residues may suggest an essential role of these amino acids for membrane association and further support membrane binding of the protein [79]. 

One advantage of using fluorescence microscopy to monitor the recruitment of proteins to membranes is that it can often observe dynamic processes that other techniques might not be able to readily observe. For example, fluorescence microscopy has been applied to monitor the dynamics of endocytosis protein clathrin on the plasma membrane in live cells over time [80]. One limitation of using fluorescently tagged proteins to study protein–membrane interactions is that the use of fluorescent tags can interfere with protein localization. In a study of yeast proteins tagged with terminal GFP tags, as many as 10% of proteins are localized to different cellular locations [81]. Additionally, there are some technical limitations of fluorescence microscopy, including a limit to the number of fluorophores that can be detected at the same time, and the spatial resolution of the microscope is a limiting factor [82]. It is important to note that in the context of studying protein–membrane interactions, fluorescence microscopy merely suggests a protein–membrane association, and any potential protein–membrane interactions need to be further confirmed using other techniques.

### 2.2. Subcellular Fractionation

Subcellular fractionation was used for some of the first experiments that suggested autophagy proteins were able to interact with membranes. In general, subcellular fractionation separates certain organelles, membrane compartments, or proteins depending on different properties such as buoyant density, charge density, and size and shape [83]. There are a wide variety of protocols that have been used depending on the cell type and cellular location of the protein of interest, but most protocols utilize sequential centrifugation to detect membrane association. Most commonly for the study of autophagy proteins in *S. cerevisiae*, spheroplasts are generated through enzymatic digestion of the cell wall. The spheroplasts are then lysed osmotically or through mechanical shear force [84,85]. Lysates are then centrifuged at a very low speed (~500× *g*) to remove large cellular debris such as unlysed or partially lysed cells and aggregates. The supernatant is then centrifuged at a faster speed (~10,000× *g*), then at high speed (~100,000× *g*) to produce low- and high-speed pellet and supernatant fractions [25,54]. The proteins in each fraction are precipitated by adding trichloroacetic acid (TCA) and analyzed by Western blot [85]. If a protein associates with membranes, it will appear in the low- or high-speed pellet fractions, while if it does not associate with membrane it will appear in the supernatant fractions [85]. 

Further density gradients can also reveal if a protein is associated with a particular organelle such as the mitochondria or peroxisomes [26]. To isolate organelles of interest, the low-speed pellet fraction from above is resuspended in buffer containing high-viscosity media such as sucrose, glycerol, Ficoll 400, and OptiPrep [25,30,79]. The resuspended pellet is then applied to a gradient of the high-viscosity media and centrifuged at high speed (170,000× *g*). Fractions are sequentially collected from the top of the gradient, TCA-precipitated, and analyzed by Western blot. Each fraction is also probed for known organelle markers to determine which organelles represent in each fraction, including Fox3 for peroxisomes and porin for mitochondria [26,83]. For example, if a protein of interest is found in a fraction of the density gradient along with porin, the protein is likely associated with mitochondrial membrane.

Differential solubilization can be used to determine if the membrane interacting-protein is integral or peripherally associated with membrane. The low- and high-speed pellet fractions are resuspended in buffer containing a reagent such as salt, Na_2_CO_3_, urea, or Triton X-100 and centrifuged [36,42]. The resuspended pellets are then centrifuged, and proteins are TCA-precipitated and analyzed by Western blot [40]. Detergents, such as Triton X-100, will dissociate both integral and peripheral membrane proteins from the membrane, thereby shifting the protein from the pellet to the supernatant fraction, while salt, pH, or denaturants will only dissociate peripherally associated proteins from the membrane [37].

The primary benefit of subcellular fractionation is that it can distinguish between an integral and peripherally associated protein if the protein is shifted from the pellet to the supernatant fraction upon treatment with detergent or salt, respectively. A major limitation of this technique is that it can detect indirect protein–membrane interactions mediated through additional protein–protein interactions. Another limitation is that it is difficult to distinguish between a protein that is membrane-bound and one that forms large aggregates found in the pellet fraction, which is where differential solubilization can be useful [44].

### 2.3. Immuno-Electron Microscopy

Immuno-electron microscopy (immuno-EM) can be applied to whole yeast cells to determine the location of a protein in cells by labeling it with specific antibodies. Immuno-EM provides a method to visualize the association of a protein with a membrane in a cellular context at ultrastructural resolution. Samples are immunolabeled with an antibody for the protein of interest and incubated with a gold-conjugated antibody, which has a high electron density and appears as dark spots in micrographs [86,87]. There are three basic approaches for immuno-EM: (1) pre-embed labeling, where the cells are labeled with the antibody for the protein of interest prior to embedding in resin; (2) post-embed labeling, where the protein of interest is labeled after the cell is sectioned and embedded in resin; and (3) cryo-sectioning, where the sample is fixed, frozen, sectioned, thawed and then labeled [88,89]. In all cases, the samples are negatively stained using heavy-atom-containing stains such as uranyl acetate or lead citrate to improve contrast of cellular structures [90]. The samples are then imaged using a transmission electron microscope. The proteins are located by the positions of the gold particles. The protein is classified as membrane-associated if it is inside an organelle or within a certain distance from a membrane, often defined as less than 50 nm [7,91,92].

One benefit of immuno-EM is that it can reveal a protein’s localization on specific regions of a membrane compartment, such as the autophagic membrane or vesicles [44]. For example, Atg9 was shown to localize to clusters of vesicular and tubular structures visualized by the presence of gold particles along the cytosolic surface of these structures [7]. One caveat to this technique is that cells must be frozen or fixated using glutaraldehyde to preserve the subcellular structures in vivo, and therefore, membrane binding dynamics are not visualized [90]. Furthermore, the fixation of biological samples can generate artifacts in the EM images such as changes in cell shape or size, or loss of visibility of membrane structures [93]. 

## 3. In Vitro Qualitative Techniques to Monitor Protein–Membrane Interactions

Once a protein has been shown to associate with membranes in cells, one possible next step is to verify that the protein directly interacts with membranes using an in vitro assay. Studyingprotein–membrane interactions using in vitro techniques requires the purification of the protein of interest and the generation of a membrane mimetic for binding. Membrane mimetics are often heterogenous with each membrane surface supporting multiple protein docking sites, complicating the determination of stoichiometries and dissociation constants [94]. This can make quantitative investigations into protein–membrane binding events challenging. As such, many in vitro membrane binding experiments are qualitative and dissociation constants, stoichiometries and binding kinetics are predominantly not determined. 

### 3.1. Membrane Mimetics

Some examples of membrane mimetics for biochemical studies include liposomes, either single- or multilamellar vesicles ranging from 50 nm (SUVs) to 1 μm (LUVs), giant unilamellar vesicles (GUVs) ranging in size from 5 μm to 100 μm, and supported lipid bilayers (SLBs), which flat membrane sheets that are attached to a surface, as shown in Figure 1 [65,75,95]. 

Thus, these different membrane mimetics can also be used to test whether membrane binding by a protein is sensitive to membrane curvature. Liposomes of different sizes can be generated to mimic the highly curved Atg9-vesicles that are 50 nm in diameter or the less curved autophagic membrane that are 300 to 900 nm in diameter [17,53]. Since SLBs are attached to a surface, they are the most stable membrane assembly and can withstand mechanical probing such as atomic force microscopy [96]. Both SLBs and GUVs are flat relative to the protein, but the lipids within GUVs are more dynamic within the membrane than SLBs since they are untethered. GUVs can be visualized using microscopy due to their large size compared to LUVs, enabling their use in resolving dynamic interactions between proteins and membranes and direct visualization of binding. 

Membrane mimetics can be generated from various lipid sources including isolated lipid mixes and synthetically generated lipids. Folch is a lipid extract from bovine brain that is commonly used for protein–membrane interaction studies [97]. However, this lipid mixture is not representative of yeast membranes [54]. Yeast polar lipid extract (YPL) also contains a high percentage of negatively charged lipids but is more representative of biological yeast membranes [53,98]. Membrane mimetics can also be produced using synthetic lipids of high purity, where the exact membrane composition can be controlled. In addition, the saturation state of the carbon tail can be selected to allow for fine control of membrane properties such as the fluidity of the membrane [10,52,55]. In these experiments, the lipid composition and morphology of the membrane mimetic should ideally resemble the biological membrane of interest. However, the exact lipid composition is not always known, and the lipids may not be evenly distributed throughout the membrane. This adds another layer of complexity when working to recreate protein–membrane interactions in vitro. 

### 3.2. Liposome Sedimentation and Floatation Assays 

Liposome sedimentation and floatation assays are widely used to study the binding and specificity of protein–membrane interactions in vitro. In liposome sedimentation assays, equal volumes of protein and liposomes are mixed, incubated together, and then centrifuged at high speed, typically greater than 80,000× *g*. The liposomes will pellet due to their size, whereas protein will only be found in the pellet upon binding to liposomes. The presence of the protein in the supernatant and pellet fractions is determined by SDS-PAGE [53,99]. Typically, a protein-only sample (with no liposomes) is utilized as a negative control to verify that the protein does not pellet under the centrifugation conditions used. 

In the liposome floatation assay, equal volumes of liposomes and protein in buffer containing a gradient medium are mixed and incubated. A density step gradient is generated by adding a reduced concentration gradient to the top of the protein–liposome mixture. For example, if the protein–liposome mixture is in a 30% sucrose buffer, the mixture is then overlaid with 25% sucrose buffer and topped with 0% sucrose buffer [100,101]. Some other choices for a density gradient include Ficoll [25,30] and Nycodenz [79]. The sample is centrifuged (greater than 160,000× *g*), fractions are collected from the top of the mixture, and the amount of protein in each fraction is determined [100]. Due to their buoyancy, liposomes and associated proteins will appear in the float fraction at the top of the gradient. If the protein does not bind to liposomes, it will not appear in the top fraction. As a negative control, liposome flotation can be performed in the presence of a detergent, where the detergent disrupts the liposomes and abolishes floatation of any membrane-bound proteins, which should then be found in the lower fractions in the density gradient [30].

An advantage that liposome sedimentation and flotation share is the ability to readily change their liposome composition (to test specificity) and size (to test curvature sensitivity) [54]. For studies of protein–membrane interactions in yeast, liposomes are typically generated with either yeast polar lipids or a composition of synthetic lipids. Between these two methods, sedimentation assays can be applied to a wider range of buffer conditions and typically require smaller volumes of both protein and liposomes. However, liposome sedimentation assays can provide a false positive for binding if the liposomes induce the protein to unfold or aggregate, causing the protein to appear in the pellet fraction. In contrast, floatation assays can be used to study proteins that aggregate or oligomerize, as aggregated proteins will not float to the same layer as liposomes [94]. A possible consideration for both assays is that some proteins can cause fission or fragmentation of vesicles. For this reason, liposomes can be labeled with a dye, such as 1,1′-Dioctadecyl-3,3,3′,3′-Tetramethylindodicarbocyanine, 4-Chlorobenzenesulfonate Salt (DiD), which will make the liposome pellet or layer visible to the naked eye if fission does not occur [94]. If protein-induced fission of vesicles occurs, the labeled liposomes will no longer be visible to the eye.

### 3.3. Recruitment to GUVs

GUVs are vesicles that are 5 μm to 100 μm in diameter and can be easily visualized by fluorescence microscopy. Unlike liposomes, which are near the size of the resolution limit of most light microscopy techniques, GUVs are large enough to monitor the recruitment of fluorescently labeled protein directly to the surface of this membrane mimetic. Fluorescently labeled GUVs are attached to glass slides and incubated with fluorescently labeled protein. Protein binding to GUVs is then monitored by colocalization using microscopy. The advantages of using GUVs are that the membranes are less curved relative to liposomes and that their large size makes them easier to resolve by light microscopy methods than LUVs [10,54,64]. Additionally, the membrane composition can be readily varied to mimic organelles or test the influence of charge on the vesicle’s surface. However, if the protein of interest prefers highly curved membranes, binding to GUVs may not occur. Another consideration is that there are many ways to generate GUVs, so methodologies for generating GUVs are quite variable [16]. 

### 3.4. Protein–Lipid Overlay Assays 

Protein–lipid overlay (PLO) assays are a method to quickly assess lipid specificity of a membrane binding protein. In PLO assays, also called PIP strip assays, different phosphoinositides (PIs) at the same concentration can be applied onto nitrocellulose or polyvinylidene fluoride (PVDF) membrane, blocked with a blocking protein such as BSA (bovine serum albumin), and then probed with a purified protein that is labeled for detection [65]. Membranes with 15 different full-length lipids are also commercially available for this assay [102]. Of note, blocking proteins may interfere with lipid–protein interactions [103]. Common methods of labeling include using a GST-tag or ^32^P-labeling, enabling detection by Western blot or radiolabeling, respectively [27,104].

One advantage of PLO assays is that the specificity of a protein for different headgroups for various PIPs or other phospholipids can be checked with relative ease and speed and require little protein (less than 0.5 μg/mL). Therefore, PLO assays can be a useful qualitative screening method for specificity. One major limitation of this technique is that PLO assays do not always report a protein’s native membrane binding specificity reliably, sometimes showing promiscuous or poor binding [102]. This is because only a single lipid is spotted densely onto the flat membrane, so binding may not be detected if the protein interacts with a different part of the lipid or the membrane itself [31,105]. 

### 3.5. Protein Structures

Both low- and high-resolution structures of autophagy proteins can provide insight into known protein–membrane interactions. These structures can also suggest possible protein–membrane binding interactions by considering structural homology to known membrane-interaction domains. There are several examples of structures of autophagy proteins that have shed light on their membrane interactions using different structural techniques.

Low-resolution negative stain or cryoEM maps can be overlaid with existing higher-resolution structures for comparison in order to see if membrane interacting regions and structural features are conserved [67,68,106]. Cryo electron tomography (cryoET) can provide structures of proteins interacting with membranes to directly visualize regions involved in the interaction [107,108]. High-resolution cryoEM structures have identified hydrophobic cavities within proteins, which are suggestive of lipid scramblase activity [69,70,71]. CryoEM structures have also enabled the direct visualization of bound phospholipids [71,109]. Finally, high-resolution crystal structures can contain phospholipid fragments coordinated by residues or by the location of sulfate ions, which are often indicators of macromolecular interactions [12,73]. 

One advantage of low- and high- resolution protein structures is that they can provide insight into how proteins interact with membrane. Moreover, the direct visualization of these interactions at atomic resolution is possible. In some instances, membrane can help stabilize regions of the protein that are not detectable with other techniques [110]. A challenge of many structural techniques is that large amounts of high-purity protein are required to determine the structure. Additionally, proteins themselves may be challenging structural targets, which is further complicated by the addition of membrane in these experiments.

## 4. Quantitative Techniques to Measure Protein–Membrane Interactions

The techniques listed above are used to qualitatively characterize protein–membrane interactions. These techniques can provide information about which lipids are important for binding and whether the curvature of the membrane plays a role in binding. However, none of the above methods can be used for quantitative measurements to determine binding affinities and stoichiometries of protein–membrane interactions. Instead, the techniques described below are often used to further characterize protein–membrane interactions in a more quantitative manner. 

### 4.1. Isothermal Titration Calorimetry

Isothermal titration calorimetry (ITC) measures the heat released or absorbed resulting from protein–membrane interactions. ITC has been utilized to characterize protein binding to either liposomes or phosphoinositide headgroups [74,104]. In ITC, an analyte is injected at a set volume and time interval from a syringe into a sample cell [111]. As binding occurs, heat is absorbed or released, and the amount of energy required to keep the sample cell at the same temperature as a reference cell containing water is recorded. In some cases, the protein is titrated into a sample cell containing liposomes [74,79,111]. Alternatively, the protein of interest can be placed into the sample cell while sequential aliquots of liposomes are injected from the syringe [104]. The average molar ratio for a protein binding to a liposome can be determined by indicating the number of binding sites per protein molecule. This is achieved by approximating the percentage of a certain phospholipid that is available on the liposome surface. Then, using the known concentrations of the phospholipid in the mixture and the concentration of the protein, the stoichiometry can be approximated. The data from an ITC experiment can also be used to determine the dissociation constant and thermodynamic properties of these binding events, including the enthalpy of binding [105]. 

One advantage of this technique is that it provides quantitative information without the need to label either the protein or the liposomes. One limitation of ITC is that it requires large amounts of protein and liposomes. In some instances, 300 μL of 100 μM protein is required [74,79]. Another consideration is that several assumptions need to be made when determining the stoichiometry and dissociation constants regarding the availability of a phospholipid for binding.

### 4.2. Surface Plasmon Resonance

Surface plasmon resonance (SPR) is another technique that has been utilized to quantitatively investigate protein–membrane interactions. In SPR, membrane mimetics are attached to the surface of a chip, and protein is then flowed over the chip at increasing concentrations [65]. As protein binds to the membrane, it changes the refractive index of the chip, resulting in a change in the resonance angle of scattered light. The change in resonance angle is directly proportional to the amount of protein bound to the immobilized membrane. Typically, the protein is injected at increasing concentrations ranging from nM to μM [31,105]. On and off rates of protein binding are calculated and used to determine the dissociation constant. There are several methods of attaching membranes to the surface of the chip. Liposomes can be flowed over the chip and fused with alkanethiol molecules on the surface to form a supported monolayer, or a bilayer can be tethered to a gold surface using thiolipids to form a tethered lipid bilayer [112,113]. In addition, intact liposomes can also be covalently attached to the surface by incorporating biotinylated lipids or lipophilic anchors that bind to a chip containing avidin or dextran, respectively [95,114,115]. 

Like ITC, SPR directly measures protein–membrane interactions without the need for fluorescent or radioactive tags. A second advantage of this technique is that smaller sample amounts are required compared to ITC, and membrane compositions can be easily varied [95,105]. Furthermore, SPR has can be used to measure both the kinetics and affinity of protein–membrane binding [31]. One consideration of using SPR is that there are many methodologies to attach the membrane to the chip, so protocols are quite variable.

## 5. Techniques to Monitor Membrane Tethering and Restructuring

Some autophagy proteins can bind to multiple membranes simultaneously, resulting in the tethering of membranes or clustering of vesicles. Other autophagy proteins can bind membranes and restructure or reshape these membranes. Lastly, other autophagy proteins have been shown to tether membranes to enable lipids to be transported between these membranes. Investigating these binding events and the specific membrane changes that are associated with them requires a different set of techniques, which are summarized in the following sections.

### 5.1. Dynamic Light Scattering to Monitor Vesicle Tethering

Dynamic light scattering (DLS) can be used to determine the size distribution of particles in solution and can also monitor changes in the size of liposomes in response to binding events. DLS measures the time dependence of scattered light; when particles undergo Brownian motion, their movement causes fluctuations in the intensity of scattered light. The scattered light is then measured at a single angle and autocorrelated to determine a diffusion coefficient, which then can be used to determine particle size [116].

In the case of membrane tethering, the effective diameter of the liposomes with and without protein is compared by DLS, where an increase in the diameter of the liposomes upon protein addition indicates tethering [53,62]. DLS can also assess the influence of liposome curvature on a protein’s tethering ability by utilizing liposomes of different sizes in this assay. Some advantages of this technique are that it is relatively fast and easy to perform that and the liposome composition and size can be varied. One disadvantage of the technique is that it can be difficult to distinguish between vesicle tethering and fusion. One way to differentiate between vesicle tethering and fusion is that proteinase K can be added, which should reverse vesicle tethering but not fusion [12].

### 5.2. Turbidity Assay to Monitor Vesicle Tethering

Turbidity assays can be used to monitor the effects of protein-induced liposome clustering. To perform this technique, liposomes are generated, and the absorbance is measured using a spectrophotometer. Protein is then added to the liposomes, and the absorbance is measured at distinct time intervals [117]. As liposomes cluster, the opacity of the sample increases, resulting in an increase in the absorbance of light [54,118]. Since the absorbance is measured at set time intervals, the kinetics of the tethering can also be determined. This technique is advantageous as a membrane tethering method due to its ease and accessibility to users; however, it is an indirect measure as it measures the change in absorbance of the entire sample that could be the result of protein aggregation instead. 

### 5.3. Bead Capture Assay for Vesicle Tethering

There are various methods that utilize fluorescently tagged lipids and measure fluorescence as a readout for membrane tethering. In this assay, two distinct liposome populations are generated with each liposome population containing a different fluorophore. One of these liposome populations is also generated with a lipid that contains an affinity tag, such as biotin-phosphatidylethanolamine (PE), that can be used for capture on an affinity resin [21]. The two liposomes are mixed in the absence and the presence of a protein of interest. The mixture is incubated with an affinity resin, such as Streptactin, that binds to one liposome species, for example containing biotin-PE, and the amount of the fluorescent liposomes tethered to the biotin liposomes is quantified using fluorescence spectrophotometry [21,52]. An advantage of the bead capture assay is that it uses fluorescence measurements that are highly sensitive and provide a direct readout of vesicle clustering [54]. One consideration when using this technique is that the size of liposomes can affect tethering results. For example, several Rab proteins were shown to tether liposomes 400 nm and 1000 nm in diameter but not 100 nm [119].

### 5.4. Fluorescence Microscopy

Fluorescence microscopy can be applied to investigate membrane tethering and membrane tubulation using fluorescently tagged lipids incorporated into GUVs. The clustering of fluorescently labeled GUVs in the presence or absence of a fluorescently tagged protein can be used to directly visualize protein-mediated membrane tethering. The protein also often accumulates at sites of GUV contact [54]. Tubulation of GUVs can be initiated by adding hypertonic GUV buffer, which allows the spherical GUVs to lose water through the permeable membrane and adjust its surface-to-volume ratio to encourage tubular structures [56]. The protein is added, and tubulation or scission of the tubules can be directly visualized. The direct visualization of tethering and the tubulation are advantages of fluorescence microscopy. However, a consideration of this method is that autophagy proteins can appear as patches on GUV surfaces, which can make quantification of binding and tethering difficult [64].

### 5.5. Electron Microscopy

Electron microscopy can be used to directly monitor liposome tethering or tubulation. To monitor the effects of proteins on liposome morphology, protein and liposomes can be incubated together and then applied to a carbon coated grid for either room temperature or cryogenic data collections. CryoEM images can be collected to examine the sample, or grids can be prepared for immuno-EM, where they are incubated with a protein-specific antibody, incubated with IgG conjugated gold particles, stained with a heavy atom stain, and imaged [51,53]. In the case of vesicle tethering, liposomes in close contact can be directly visualized, whereas in vesicle tubulation, lengths of tubules can be directly measured [57]. Furthermore, immuno-EM can be used to visualize the enrichment of a protein at the junction of two liposomes [51]. A major advantage of this technique is that it is used to directly visualize and quantify vesicle clustering and tubule width [57,58,59,120]. One limitation of using EM to monitor membranes is that it requires the manual analysis of membrane structures, thereby limiting the throughput and potentially introducing bias by the limited field of view [121]. 

### 5.6. Lipid Transfer Assays 

Some autophagy membrane tethers have also been shown to transfer lipids between membranes, such as Atg2 and Atg8. First described by Struck et al., 1981, the lipid mixing assay has been used to test for lipid transfer activity in proteins. Lipid transfer assays rely on fluorescence resonance energy transfer (FRET), in which an excited donor fluorophore transfers its energy to a nearby acceptor fluorophore in a distance-dependent manner due to the overlap in their emission and excitation spectra, respectively [122]. Thus, FRET is an accurate measure of proximity of macromolecules. In this assay, two distinct liposome populations are generated, one containing *N*-(7-Nitrobenz-2-Oxa-1,3-Diazol-4-yl)-1,2-Dihexadecanoyl-*sn*-Glycero-3 (NBD)- and Rhodamine- labeled lipids, and the other lacking any fluorescent lipids. The energy transfer from liposomes containing NBD to Rhodamine is measured, each conjugated to PE. If both conjugated dyes are in the same liposome, the NBD fluorescence is quenched by the Rhodamine. However, if lipid transfer occurs between the labeled and unlabeled liposomes, the distance between the two dyes on the membrane will increase, and the NBD fluorescence will be increase as a result of reduced quenching from the Rhodamine. Thus, the fluorescence of NBD can be monitored as a readout for lipid transfer activity [123]. 

A major advantage of this assay is that there is high sensitivity afforded by the fluorophores and high versatility afforded by changing the liposome composition. However, one experimental note is that lipid transfer activity sometimes requires the addition of ATP [12,51]. Additionally, lipid transfer rates measured in vitro may be slower than rates found in vivo [16]. Another consideration is that this technique does not distinguish between lipid transfer from vesicle fusion, so other techniques such as EM can be used in combination to differentiate between these two events.

## 6. Autophagy Protein–Membrane Interaction Experiments

### 6.1. Atg1 Initiation Complex 

The Atg1 initiation complex is comprised of five core autophagy proteins (Atg1-Atg13-Atg17-Atg31-Atg29) and mediates autophagy initiation [4,53,124]. This complex assembles at the phagophore assembly site (PAS) and recruits Atg9-containing vesicles derived from the Golgi, proposed to be the initial membrane source for the isolation membrane [8].

The membrane association of Atg1 was first suggested by subcellular fractionation, where the protein was found in the high-speed pellet fraction and shifted to the supernatant fraction when Triton X-100 was added [35]. Interestingly, the protein’s membrane association was dependent on its Atg8-interaction motif (AIM), as mutations of this region disrupted its membrane association. These results suggested that the interaction of Atg1 with membranes may be indirect and require binding to Atg8. However, the C-terminus of the mammalian homolog of Atg1, ULK1, had previously been shown to bind membranes [125], which conversely suggested that Atg1 may bind membranes directly. Liposome sedimentation assays were performed with the C-terminal MIT domains of Atg1 using Folch, YPL, and different synthetic compositions of various sizes, all showing a preference for SUVs of high curvature (20–30 nm) that are enriched for phosphatidylinositol (PI) or phosphatidylinositol-3-phosphate (PI3P) [53]. The MIT domains of Atg1 was also shown to act as a vesicle tether by both DLS and a bead capture assay [52].

Atg13 was first identified to interact with membranes using in vitro liposome sedimentation assays. Purified Atg13 bound to highly curved liposomes generated from YPL [53]. Next, binding between Atg13’s C-terminal intrinsically disordered region and liposomes were tested. Atg13 bound to Folch-derived liposomes in a salt-dependent manner, indicating that the interaction is mediated through electrostatics [55]. Upon further investigation, Atg13 bound to synthetic liposomes with high negative charge, where an increase in PS from 19% to 42.5% PS led to increased binding. Atg13 binding was further enhanced upon addition of PI3P. Lastly, two phenylalanine containing binding sites were identified, where aromatic residues are postulated to be inserted into the membrane upon binding [55].

The crystal structure of Atg17-31-29 revealed that the Atg17 monomer has a crescent shape, with a radius of curvature of 100 Å, which is consistent with the curvature of an Atg9 vesicle [52]. Therefore, to test the hypothesis that Atg17 binds membranes, liposome sedimentation assays were performed with the Atg17-31-29 complex and different-sized liposomes made from Folch, in which no binding was detected. However, Atg17 was shown to bind and tether Atg9-proteoliposomes in vitro using floatation assays and DLS, mediated through its interaction with Atg9 [53].

Liposome floatation assays with the Atg1 pentameric complex containing Atg1-Atg13-Atg17-Atg31-Atg29 and YPL-derived liposomes of various size were performed, where the complex retained a preference for small liposomes [53]. The interaction between Atg9-proteoliposomes and Atg17 was shown to be inhibited by Atg31-Atg29 in a floatation assay, which could be negated by the addition of Atg1-13. Lastly, this study utilized CryoEM to directly visualize multiple Atg9-proteoliposomes tethered by the Atg1 complex.

### 6.2. PI 3-Kinase Complex

PI3P is an essential phospholipid in autophagy, as it enables the recruitment of the Atg2-Atg18 complex to the PAS by binding directly to PI3P. In yeast, Vps34 is the sole PI 3-kinase (PI3K) responsible for the phosphorylation of PI to generate PI3P [43]. Vps34 is a component of two separate tetrameric PI3K complexes: PI3K complex I is involved in autophagy and will be discussed in this review, while PI3K complex II is involved in endocytosis [126]. Both complexes comprise Vps34, Vps15, and Vps30/Atg6 with either Atg14 or Vps38 for complex I or II, respectively [43]. Some studies have investigated the membrane interactions of individual proteins of complex I from yeast. However, there are limited studies involving the complete yeast PI3K complex I, so we will also review what is known from other model systems.

In yeast, Vps15 and Vps34 were initially probed for membrane association using subcellular fractionation. This experiment revealed that Vps15 was membrane-associated, while Vps34 was associated with membranes in a Vps15-dependent manner [50]. Atg14 was also found to be peripherally membrane-associated by subcellular fractionation [42]. The treatment of the Atg14-associated pellet with Na_2_CO_3_ shifted the protein to the supernatant, suggesting that it is peripherally associated. By subcellular localization, Atg14 was also shown to localize to the PAS and the vacuolar membrane [28,29]. Additionally, Vps30/Atg6 was shown to be peripherally membrane-associated by subcellular fractionation; however, the deletion of Vps34 and Vps15 shifted the protein to the supernatant [41,42,43]. 

The complete PI3K complex I was purified and found to be associated with and active in a lipid kinase assay on highly curved SUVs in vitro but not with GUVs [127]. There are no structures to date of yeast PI3K complex I. However, the crystal structure of the yeast PI3K complex II shows that it forms a Y-shape, where one arm includes Atg6/Vps30 and Vps38, while the other arm includes Vps34 and Vps15 [127]. Hydrogen-deuterium exchange (HDX) mass spectrometry was used to identify which regions within yeast complex II are involved in membrane binding. HDX analysis revealed that Vps30/Atg6 beta-alpha-repeated, autophagy specific (BARA) domain with an aromatic finger had a decrease in solvent exchange, suggesting that this region interacts with the membrane. There are also changes in solvent exchange at the Vps34 activation loop in the other arm, which suggest that the complex II-membrane interaction is mediated through both arms of the Y-shaped structure [127].

The yeast complex II crystal structure is consistent with the low-resolution EM structure of the human class I PI3-kinase complex I, suggesting that structural features of these complexes are conserved [106]. Recently, a cryoET structure of human complex II bound to the membrane was determined, which could provide insight into the yeast complex I–membrane interaction. The cryoET structure revealed that an adaptor arm of BECN1 (Atg6 in yeast) mediates the highly dynamic complex’s interaction with the membrane, which is consistent with the aromatic finger identified in the yeast complex II HDX studies [108].

### 6.3. Transmembrane Protein Atg9 

Atg9 is the only known transmembrane protein of the core autophagy machinery. Atg9-containing vesicles are proposed to be the initial membrane source at the PAS for the forming autophagosome [7,8]. These single membrane vesicles are highly mobile in the cytoplasm and are proposed to be derived from the Golgi [7,128].

Hydrophobicity analysis initially predicted that Atg9 consists of six to eight membrane-spanning regions [25]. To test this, subcellular fractionation was employed, where Atg9 was found to pellet with membrane compartments distinct from known organelles [77]. Only detergent was able to be extracted Atg9 from the pellet fraction, indicating that it is an integral membrane protein [26]. The subcellular fractionation of Atg9 also revealed that Atg9 co-migrated with autophagosomal components, where it was sensitive to proteinase K; thus, Atg9 is found in the outer autophagosomal membrane [8]. Atg9 has a unique cellular localization featuring puncta near the vacuole and the mitochondria, and several smaller cytosolic puncta were visualized with fluorescence microscopy [25,124]. 

Immuno-EM was used to directly visualize Atg9 and its association with the membrane. Atg9-containing vesicles were approximately 30 nm in diameter and found as clusters [7]. Furthermore, Atg9-containing tubules were seen at the PAS, which likely corresponds to membrane sheets. Negative-stain EM further confirmed that Atg9-containing vesicles are single membrane vesicles that are 30–60 nm in diameter [8]. 

Recent work has shed light on the function of Atg9 as a lipid scramblase. A dithionite assay was used to test if Atg9 can transfer lipids from the outer to inner leaflet of the isolation membrane [70]. In this assay, liposomes containing NBD are treated with dithionite, a membrane impermeable reducing agent, which reduces and irreversibly quenches NBD fluorescence on the outer leaflet, resulting in about 50% quenching of the fluorescence signal [129]. In the presence of a lipid scramblase, some NBD-labeled lipids are transported from the inner to the outer leaflet of liposomes, resulting in an increased amount of quenching. When Atg9 was added to these liposomes, greater than 50% quenching occurred in a concentration-dependent manner, indicating that Atg9 functions as a lipid scramblase in vitro [70,71]. Atg9 was also able to transport PI3P from the outer leaflet to the inner leaflet of liposomes in vitro, where the PI3P distribution was directly visualized using electron microscopy [70]. 

The cryoEM structure of yeast Atg9 features a homotrimer, where each monomer has four transmembrane helices, two helices laterally embedded in the membrane, and two large cytosolic termini [70]. Each monomer contains a small pore that accommodates amphipathic molecules, and the trimer assembly features a large hydrophilic pore, by which it likely scrambles lipids. This structure is consistent with the high-resolution human cryoEM structures, which featured a lipid in the hydrophilic cavity [69,71]. Mutations to the large hydrophilic core resulted in reduced scramblase activity, further corroborating the role of this pore. The 7.8 Å *Aradopsis thaliana* cryoEM map is also consistent, demonstrating that Atg9 structural features are conserved [68]. 

### 6.4. Atg2–Atg18 Complex 

The Atg2–Atg18 complex has shown to be involved in phagophore expansion [10,20,32]. Atg18 belongs to the beta-propellers that bind phosphoinositides (PROPPINS) family that have specificity for PI3P and PI3,5P [130]. While Atg18 is a core autophagy protein, its two paralogs Atg21 and Hsv2 only function in the cytoplasm-to-vacuole (Cvt) pathway and microautophagy of the nucleus, respectively [130,131]. 

Atg2 was first suggested to bind membranes via subcellular fractionation, where it was found in both the soluble and pellet fractions and associated with the endoplasmic reticulum (ER) [21,36,37]. Treatment of the Atg2-associated pellet with salt, Na_2_CO_3_, and urea could increase its solubility, suggesting that it is peripherally associated. However, it was unclear if Atg2 directly interacted with the membrane, or if this interaction was mediated through Atg18. To test this, liposome floatation assays with Atg2 and an Atg18-mutant unable to bind membrane revealed that Atg2 directly bound to the PI3P-containing membrane and preferred small liposomes [10,21]. Microscopy was performed for Atg2 with fluorescent GUVs, where Atg2 was only recruited to PI3P-containing membrane [10]. Furthermore, Atg2 formed patches along the GUV surface that were also enriched for the fluorescent dye Atto550-1,2-Bis(diphenylphosphine)ethane (DPPE) [21]. This fluorescent dye partitions into regions of looser lipid packing in the membrane; thus, it was hypothesized that Atg2 bound to membrane regions that had lipid packing defects [132]. To test this hypothesis, inducers of packing defects such as ergosterol and inhibitors of packing defects such as PE were tested, confirming that Atg2 associates at regions of lipid packing defects [10]. 

By fluorescence microscopy, the N-terminus of Atg2 was found to target the ER, while the C-terminal amphipathic helix targets the PAS, raising the possibility that Atg2 spans two membranes [21]. Thus, Atg2’s ability to tether the membrane was tested using the fluorescent bead assay, lipid mixing assay, and DLS, confirming that Atg2 is indeed able to tether and transfer lipids between liposomes via its N- and C- termini [12,21]. Further corroborating a role in lipid transfer, the crystal structure of the N-terminal region of Atg2 (Atg2^NR^) contained a Chorein domain, which featured a large hydrophobic cavity in its center, reminiscent of lipid transfer proteins [12]. The co-complex of Atg2^NR^ with the lipid PE was pursued, where the electron density of the acyl chain and the phosphate of the headgroup within the hydrophobic cavity were observed. This revealed molecular details of the Atg2–PE interaction and that Atg2 has little specificity for the headgroup.

Atg18 was found to be partially associated with membrane by subcellular fractionation experiments, which was disrupted by pH or by detergent, suggesting that it is peripherally associated [30]. Fluorescence microscopy also showed that Atg18 colocalizes along the vacuolar rim, endosomes, and autophagic membranes, further supporting its membrane interaction [28,30,31,32]. The lipid binding specificity of Atg18 has been controversial, with some discrepancies in results from various methods. Many techniques have been used to demonstrate that Atg18 binds specifically to PI3P and/or PI3,5P via a conserved FRRG motif including protein-overlay assays, liposome sedimentation, liposome floatation, and surface plasmon resonance, with differing results [21,28,31,56]. However, crystal structures of Atg18 and its paralog Hsv2 revealed two phosphoinositide binding sites mediated by each of the arginine residues in the FRRG motif, by which it is likely able to bind both PI3P and PI3,5P [72,73]. To further characterize Atg18 interactions with PI3P and PI3,5P, GUVs with these lipids and 1% rhodamine-PE were generated and imaged with confocal microscopy. Interestingly, when Atg18 was added to the GUVs, tubulation and scission into smaller vesicles was induced [56]. 

The localization of the Atg2-Atg18 complex to the isolation membrane is mediated by binding PI3P, as shown by fluorescence microscopy [32]. The negative stain EM structure of yeast Atg2-Atg18 reveals that it is an extended rod shape with a distinct bead-like feature at one end, much like the mammalian structure [67]. Atg2 and Atg9 have been shown to directly interact, and microscopy suggests that they form the contact site between the expanding isolation membrane and the ER, directly mediating membrane expansion [10]. 

### 6.5. The Atg12~Atg5 Conjugation System

Autophagy requires two ubiquitin-like conjugation systems. The first is the Atg12~Atg5 conjugation system primarily involved in expansion of the isolation membrane, in which ubiquitin-like Atg12 is activated by E1-like Atg7, transferred to E2-like Atg10, and then covalently conjugated to Atg5 [39]. Atg12~Atg5 then associates with dimeric Atg16 to form a multimeric membrane-associated complex [133]. Multiple members of this complex, including Atg5 and Atg16, have been implicated in membrane binding. However, Atg10 binding to the membrane has not been characterized.

Subcellular localization by fluorescence microscopy first suggested an association of Atg7 with membrane, where it formed perivacuolar punctate structures upon nitrogen starvation, which appear to be dependent on Atg12 [24]. These findings were confirmed by subcellular fractionation, where Atg7 shifted from primarily in the supernatant fraction to the pellet fraction in starved cells. However, Atg7 was unable to form puncta in *atg12*Δ cells, indicating that its membrane interaction may be indirect.

Atg5 was suggested to be membrane-associated using subcellular fractionation [39,40]. Liposome sedimentation assays further confirmed the binding of Atg5 to Folch liposomes in vitro, where the addition of Atg12 reduced the binding, which was then restored upon further addition of Atg16 [54]. This suggests that Atg12 inhibits Atg5-membrane binding, but then Atg16 alleviates this inhibition. Atg12 membrane binding alone has not been tested due to an inability to purify the protein [54].

Atg16 is required to localize Atg5-Atg12 to the PAS, suggesting a role in membrane binding [134]. Using a liposome sedimentation assay, Atg16 alone was not shown to bind to Folch liposomes but promoted binding of the Atg5-Atg12 complex [54]. Atg16 was found in the membrane fraction in a subcellular fractionation in multiple knockout (MKO) cells, which lack 24 genes required for autophagy [46,135]. Further investigation of Atg16 by liposome sedimentation assays with YPL liposomes identified an amphipathic helix that mediates its interaction with membranes with little preference for curvature [46]. 

Liposome sedimentation experiments with Folch, PE/PI, PE/PI3P liposomes, and either Atg12~Atg5-Atg16 or Atg5-Atg16 both exhibited highest binding to charged Folch liposomes, suggesting that the binding is charge-dependent [54]. The insertion of diacylglycerol (DAG), which lacks a headgroup, promoted binding, suggesting that the complex inserts hydrophobically into membranes. The Atg12~Atg5-Atg16 complex formed large clusters on GUVs and was found at GUV–GUV contacts. This conjugation complex was also found to tether liposomes in vitro using fluorescently labeled GUVs and a turbidity assay [54]. 

### 6.6. The Atg8~PE Conjugation System

The second conjugation complex in autophagy results in the covalent attachment of the ubiquitin-like protein Atg8 to the lipid PE. This ubiquitin-like cascade involves cleavage of the C-terminus of Atg8 by Atg4, followed by the activation by Atg7, transfer to E2-like Atg3, and conjugation by the previously discussed Atg12~Atg5-Atg16 complex [61,77,136]. Atg8~PE conjugation is required for membrane tethering and hemifusion during autophagy. Subcellular fractionation of Atg4 found it to be distributed only in the soluble fraction. Further characterization found that the association of Atg4 with the PAS is dependent on Atg8 binding [77,137]. 

Atg3 was originally characterized as a soluble protein, as subcellular fractionation showed that it was found in the supernatant fraction [77]. However, further studies found that Atg3 localized uniformly to the surface of the isolation membrane by microscopy, suggesting a membrane association [22,23]. Floatation assays revealed that Atg3 directly bound to liposomes containing greater than 50% PE and that acetylation of Atg3 improved its membrane binding [60,61]. Based on these findings, it is thought that the acetylation of Atg3 enhances its recruitment to membrane and thus Atg8~PE conjugation.

Atg8 was found in the membrane pellet upon subcellular fractionation, and detergent solubilization drove the protein into the supernatant fraction, indicating that it was membrane-bound [44,45]. Further analysis by fluorescence microscopy and immuno-EM showed that Atg8 localized to isolation membranes [44]. To investigate the membrane association of Atg8, the purified protein was analyzed by mass spectrometry to determine its mass [136]. This analysis revealed that Atg8 was covalently bound to a molecule with a mass consistent with PE. It was thereby concluded that the C-terminus of Atg8 is anchored to membranes by its association with PE. 

In vitro Atg8~PE conjugation requires Atg7, Atg3, ATP, and PE-containing liposomes [63,138,139]. Using this system, Atg8 was shown to mediate membrane tethering, as evidenced by accumulation of the protein at GUV contact sites [51,54,63]. This tethering was further characterized using turbidity, light microscopy to visualize vesicle clusters, and DLS [51]. To test if Atg8 mediates membrane fusion in addition to tethering, a lipid mixing assay was employed in which the level of fusion was observed to increase in an Atg8-dose-dependent manner. Immuno-EM revealed that Atg8 mediates hemifusion, which is the intermediate fusion of the membrane where the outer leaflets merge but inner leaflets are intact, and not fusion, since it was enriched at the junction between liposomes [51]. 

The Atg12~Atg5-Atg16 complex enhances Atg8~PE conjugation in vitro [54,138,140]. Interestingly, when the Atg12~Atg5-Atg16 complex was added to Atg8~PE-conjugated GUVs, deformations in the membrane occurred, and the complex formed an immobile protein layer on membranes [64]. To further characterize organization of the Atg12~Atg5-Atg16 complex on membranes, the complex was assembled on supported lipid bilayers and studied using atomic force microscopy. Interestingly, it was found that Atg8-PE and the Atg12~Atg5-Atg16 conjugation complex form a mesh-like scaffold on the membrane, but it has yet to be determined if this scaffold occurs in cells [64].

### 6.7. Additional Autophagy Factors 

Other proteins beyond the core autophagy proteins have also been shown to interact with membranes. These include Vac8, which is required to tether the autophagosome to the vacuolar membrane; Atg23 and Atg27, which are involved in Atg9 vesicle biogenesis; the selective autophagy scaffolding protein Atg11; and Atg20 and Atg24, which are sorting nexin proteins. All of these proteins are required for Cvt and efficient selective autophagy. 

### 6.8. Vacuolar Protein Vac8

Vac8 is anchored to the vacuolar membrane via the myristoylation of a glycine residue and palmitoylation of three cysteine residues in its N-terminus [33]. Vac8 tethers the Atg1 complex to the vacuole for initiation of autophagosome biogenesis. Additionally, Vac8 is involved in both nonselective and selective autophagy, where it binds to the C-terminus of Atg13 and the scaffolding protein Atg11, thereby facilitating the assembly of the PAS at the vacuole [34,141,142]. 

Vac8 was first shown to be membrane-associated by subcellular fractionation. Further fractionation and fluorescence microscopy experiments revealed that Vac8 binds to the vacuolar membrane via its N-terminus [33]. Electron microscopy studies revealed that the deletion of Vac8 caused a reduction in autophagosomal size and number and vacuolar fusion, suggesting that Vac8 plays a role in this stage [34]. To test this idea, an in vitro autophagosome-vacuole fusion assay was performed. In this assay, fluorescently labeled vacuoles were isolated from cells lacking the vacuolar lipase Atg15, and isolated GFP-Atg8-containing autophagosomes were mixed together in the presence of ATP. Fluorescence microscopy was then used to quantify the number of GFP-positive vacuoles, indicating in vitro fusion [5,34]. When testing the role of Vac8 in this process, vacuoles were isolated from cells lacking both Vac8 and Atg15. No GFP was detected in the vacuole, indicating that fusion was blocked [34]. These results indicate that Vac8 is involved in autophagosomal-vacuolar fusion. 

### 6.9. Atg11

Atg11 is a scaffolding protein that bridges the expanding isolation membrane to the cargo in selective autophagy, playing an analogous role to Atg17’s function in nonselective autophagy. Previously, Atg17 had been shown to tether Atg9-proteoliposomes [53]; thus, it was hypothesized that Atg11 may also bind and tether in the same manner. To test this, floatation assays were performed with purified Atg11- and Atg9-proteoliposomes. Atg11 only bound to liposomes containing Atg9, indicating that its interaction with membrane is indirectly mediated through Atg9. DLS was used to test if Atg11 could tether vesicles; monomeric Atg11 was unable to tether liposomes, while Atg11^ΔC^ could tether Atg9-proteoliposomes. Interestingly, the addition of the mitophagy receptor Atg32 containing a phosphomimetic mutation was found to activate Atg11-mediated tethering of Atg9-proteoliposomes visualized by cryoEM.

### 6.10. Atg23 and Atg27

The peripheral membrane protein Atg23 and transmembrane protein Atg27 are required for the efficient recruitment of Atg9 to the PAS [124]. Atg23 and Atg27 have also been implicated in the organization of Atg9-containing vesicles at the Golgi prior to vesicular transport to the PAS [143]. Subcellular localization of these three proteins differs from other autophagy proteins that only localize to the PAS, where these proteins form multiple punctate structures throughout the cell, including near the mitochondria and Golgi [76]. Furthermore, these three proteins’ trafficking is interdependent on each other [143]. 

Subcellular fractionation of Atg23 revealed that some of the protein was found in the membrane fractions; however, it was unclear if this interaction was direct or mediated through Atg9 [47]. To distinguish between these two possibilities, in vivo subcellular fractionation in MKO cells and in vitro liposome sedimentation assays revealed that Atg23 directly interacts with YPL liposomes in an electrostatic manner [48]. The elongated shape of Atg23 suggested that it may function as a membrane tether. To test this idea, Atg23 was recruited to fluorescent GUVs, where the protein was able to further recruit small liposomes to the GUV, indicating that Atg23 functions as a vesicle tether [48]. 

Atg27 was first suggested as a single-pass transmembrane and PI3P-binding protein via a positively charged region using a protein–lipid overlay assay [66]. Subcellular fractionation further supported that Atg27 is a transmembrane protein, as it was found in the membrane fraction [144]. To determine the orientation of the protein in membrane, the Atg27 was C-terminally tagged with HA and the fractionation pellets were treated with detergent and proteinase K. The C-terminal tag was cleaved by the protease in the presence and absence of detergent, indicating that the C-terminus is cytosolic. The PI3P binding of Atg27 remains to be further explored.

### 6.11. Atg20 and Atg24

Atg20 and Atg24 are sorting nexins proteins that contain an N-terminal Phox homology (PX) domain for PI3P recognition and a C-terminal Bin/Amphiphysin/Rvs (BAR) domain, which dimerizes to form a crescent shape [57]. Atg24 forms a homodimer, while Atg20 forms a heterodimer with Atg24. 

Atg20 was predicted to have a PX domain, which typically has specificity for PI3P. To test this, protein–lipid overlay assays were performed using the PX domain of Atg20, demonstrating a preference for PI3P [27,65]. Using SPR, Atg20 exhibited poor binding to PI3P [65]. The subcellular localization of Atg20 revealed that the protein is diffusely cytosolic and formed punctate structures near the vacuole by fluorescence microscopy, where the deletion of the PI-3 kinase Vps34 disrupted its localization [27]. Further confirmation of membrane association was revealed by subcellular fractionation experiments; Na_2_CO_3_ could release the protein from pellet fractions, suggesting that it is peripherally associated [27]. Within the BAR domain of Atg20, a noncanonical GAP region (residues 298-358) was identified and is suggested to be an amphipathic helix [57]. 

Atg24 was also predicted to bind the membrane through a putative PX domain [65]. To confirm the predicted specificity of Atg24, a protein–lipid overlay assay and SPR were used to characterize the protein, revealing that it prefers PI3P and PI3,5P to a lesser extent [27,65]. Subcellular fractionation was performed to test if Atg24 was capable of binding to the membrane. Atg24 was found both in the supernatant and membrane fractions in these experiments. Na_2_CO_3_ was able to remove Atg24 from the membrane fractions, suggesting that it was a peripheral membrane protein [27]. By fluorescence microscopy, it was found to have a diffuse cytosolic localization along with punctate structures near the vacuole. Like canonical BAR-domain containing proteins, Atg24 forms a homodimer [27]. Liposome sedimentation assays were used to test if the Atg24 homodimer binds to the membrane, revealing that it binds to PI3P- and PI3,5P-containing liposomes [59]. 

Atg20 and Atg24 have been shown to form a heterodimeric complex [27]. Atg20-Atg24 was shown to directly bind to negatively charged Folch and PS-containing liposomes using liposome sedimentation assays and preferentially bound to larger liposomes [57,58]. Point mutants to the GAP region within Atg20 greatly reduced membrane binding of the complex by liposome sedimentation assays, implicating this helix in membrane interaction [57]. Atg20–Atg24 was also shown to preferentially bind to liposomes with PI3P and PI3,5P and liposomes with an autophagosome-like composition [59]. 

Atg20 and Atg24 are also predicted to contain BAR domains, which are able to deform membrane and coat endosome-derived tubules [145]. To test this idea, live-cell time-lapse fluorescence microscopy was used to show that Atg20 and Atg24 are independently able to coat tubules [146]. Negative stain EM was used to show that Atg20–Atg24 mediated tubulation of liposomes, where tubulation increased with PS concentration [57,58]. A recent study compared the tubulation of the Atg24–Atg24 homodimer and the Atg20–Atg24 heterodimer using negative-stain EM [59]. These experiments revealed that Atg20–Atg24 can tubulate 5% PI3P containing and autophagosomal-like membranes, while Atg24–tg24 cannot.

## 7. Conclusions 

To gain insight into the molecular mechanisms of the complex membrane rearrangements that occur during autophagy, it is essential to understand which autophagy proteins bind membranes, how these interactions are mediated, and how they interact temporally. In this review, we have discussed the techniques that have been utilized to study protein–membrane interactions in yeast autophagy and tracked their application to each autophagy complex. A summary of the membrane interactions in yeast autophagy can be found in Figure 2. Mass spectrometry has revealed the phospholipid composition of the autophagic membrane. To do this, autophagic membranes were co-immunoprecipitated using 2xGFP-Atg8 and analyzed by phospholipidomics. Their results indicate that autophagic membrane is composed of 38% PC, 19% PE, and 37% PI, with most fatty acids found to be highly unsaturated [13]. Addressing this previously longstanding challenge of the autophagy field provides significant insight into membrane properties involved in these interactions. Determination of this lipid composition will continue to help shape the design of future protein–membrane interaction and autophagy membrane reconstitution experiments. 

## Figures and Tables

**Figure 1 cells-11-01876-f001:**
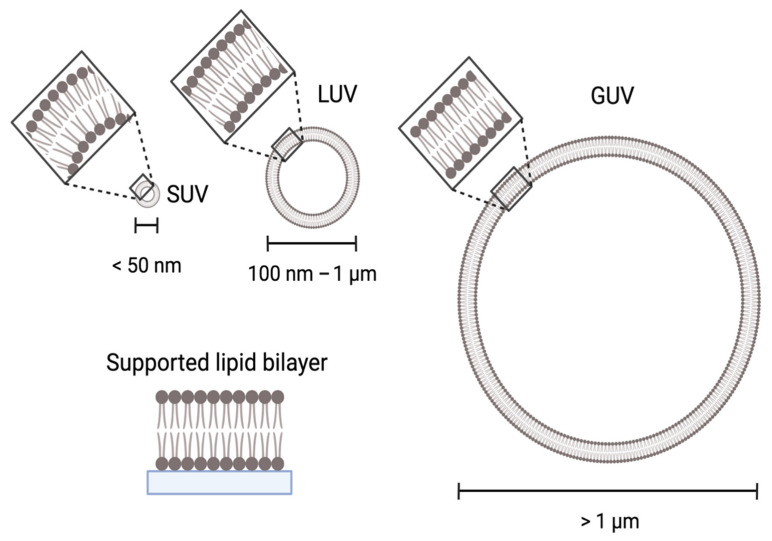
Types of synthetic bilayers of various sizes and curvature that are used in in vitro experiments. The type of synthetic bilayers varies depending on the method. For example, SUVs and LUVs are often used in liposome sedimentation or floatation assays, GUVs are often used in fluorescence microscopy, and SLBs are used in SPR. In all cases, the composition of the synthetic bilayer can be generated using lipid mixtures, such as Folch or YPL, or with synthetic lipids. Created with BioRender.com (accessed on 8 June 2022).

**Figure 2 cells-11-01876-f002:**
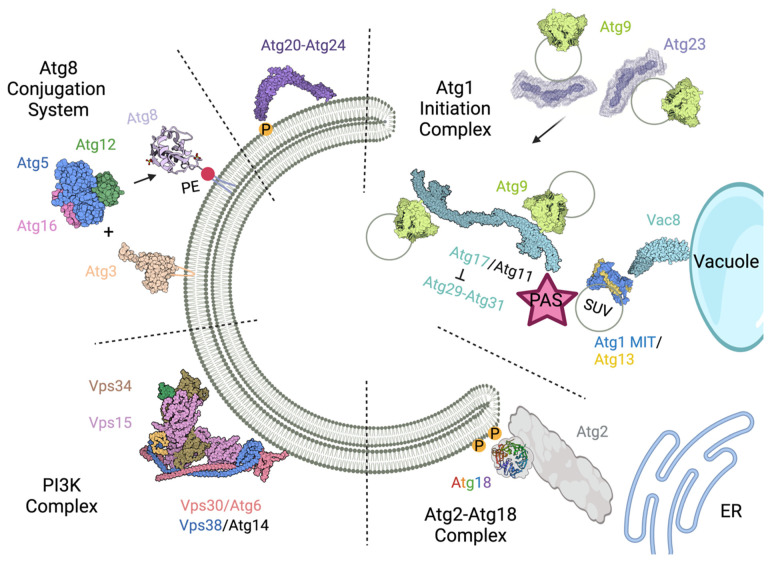
Summary of membrane interactions in yeast autophagy. Atg1-initiation complex: Atg17 (and Atg11 in selective autophagy) tethers Atg9-vesicles, which is inhibited by Atg29-Atg31 binding. Vacuolar protein Vac8 binds to the initiation complex to tether the PAS to the vacuole. The C-terminal MIT domain of Atg1 and the disordered region of Atg13 bind to SUVs. Atg2-Atg18 complex: Atg18 binds to PI3P, while Atg2 transfers lipids between the ER and the isolation membrane. PI3K complex: The Y-shaped complex binds membrane via the Vps34 activation loop and the Vps30/Atg6 aromatic finger in the BARA domain to phosphorylate PI to PI3P. Atg8-conjugation system: Atg3 and the Atg12~Atg5-Atg16 complex all bind to membrane in vitro and are involved in Atg8-lipidation. The C terminus of Atg8 is covalently attached to PE in the isolation membrane. Additional autophagy factors Atg20 and Atg24 form a heterodimer that binds to PI3P and the isolation membrane in vitro. Atg23 can bind to and tether membrane to organize transmembrane Atg9-containing vesicles. The following PBD IDs are used in this figure Atg9 (7JLP), Atg1/13 2014 (4P1N), Atg17 (4HPQ), PI3K (5DFZ), Atg18 (6KYB), Atg8 (2ZPN), Atg3 (2DYT), Atg12~Atg5-Atg16 (3W1S), and Vac8 (6KBM). Created with BioRender.com.

**Table 1 cells-11-01876-t001:** Summary of autophagy proteins and their functions.

Yeast Protein	Mammalian Counterpart	Function in Autophagy
Atg1 complex (Atg1, Atg13, Atg17, Atg29, Atg31)	ULK1 complex (ULK1, ATG13, FIP200, ATG101)	Initiates autophagosome formation, recruits and tethers of Atg9 vesicles, and phosphorylation of autophagy proteins.
PI3K complex I (Atg14, Vps34, Atg6/Vps30, Vps15)	PI3K complex I (ATG14L, VPS34, BECN1, VPS15)	Phosphorylation of PI to generate PI3P at the PAS for protein recruitment.
Atg9	ATG9A	Transmembrane protein in single membrane vesicles that supply the initial membrane source for the autophagic membrane. It additionally functions as a lipid scramblase on the isolation membrane.
Atg2–Atg18 complex	ATG2A-WIPI complex	Binds PI3P for PAS targeting; tethering and lipid transport from ER to the isolation membrane.
Atg12~Atg5 conjugation complex (Atg7, Atg10, Atg12, Atg5, Atg16)	ATG12~ATG5 conjugation complex (ATG7, ATG10, ATG12, ATG5, ATG16L1)	Ubiquitin-like cascade to conjugate Atg12 to Atg5.
Atg8~PE conjugation complex (Atg4, Atg7, Atg3, Atg8)	LC3~PE conjugation complex (ATG4, ATG7, ATG3, LC3/GABARAP)	Results in conjugation of Atg8 to PE. Involved in expansion and closure of isolation membrane.
Vacuolar protein Vac8	Unknown	Tethers the Atg1 complex to the vacuolar membrane for autophagosome biogenesis. Required for autophagosome-vacuole fusion.
Selective autophagy scaffolding protein: Atg11	FIP200	Scaffold between isolation membrane and selective autophagy receptors.
Accessory factors: Atg23, Atg27	Unknown	Generation of Atg9 vesicles, recruitment of vesicles to the PAS.
Accessory factors: Atg20–Atg24	Unknown	Sorting nexins.

**Table 2 cells-11-01876-t002:** Summary of methods used to characterize protein–membrane interactions in yeast autophagy.

Technique	Protein	Reference
**In Cell Techniques**		
Subcellular Localization by Fluorescence Microscopy	Atg2	[20,21]
	Atg3	[22,23]
	Atg7	[24]
	Atg9	[25,26]
	Atg13	[27]
	Atg14	[28,29]
	Atg18	[28,30,31,32]
	Atg20	[27]
	Vac8	[33,34]
Subcellular Fractionation	Atg1	[35]
	Atg2	[21,36,37]
	Atg3	[38]
	Atg5	[39,40]
	Atg6/Vps30	[41,42,43]
	Atg7	[24]
	Atg8	[44,45]
	Atg9	[8,25,26,38]
	Atg14	[42]
	Atg16	[46]
	Atg18	[30]
	Atg20	[27]
	Atg23	[47,48]
	Atg24	[27]
	Atg27	[49]
	Vac8	[33]
	Vps15	[50]
Immuno-EM	Atg8	[51]
	Atg9	[7]
**In vitro Qualitative Techniques**		
Liposome Sedimentation Assay	Atg1	[52,53]
	Atg5	[54]
	Atg13	[53,55]
	Atg16	[46,54]
	Atg17	[52]
	Atg18	[56]
	Atg20-Atg24	[57,58,59]
	Atg23	[48]
	Atg24	[59]
Liposome Floatation Assays	Atg1-Atg13-Atg17-Atg31-Atg29	[53]
	Atg2	[10,21]
	Atg3	[60,61]
	Atg11	[62]
Recruitment to GUVs	Atg2	[10]
	Atg8	[51,54,63]
	Atg12~Atg5-Atg16	[54,64]
	Atg18	[56]
	Atg23	[48]
Protein–Lipid Overlay Assay	Atg13	[27]
	Atg18	[28,31]
	Atg20	[27,65]
	Atg24	[65]
	Atg27	[66]
Protein Structures	Atg2	[12]
	Atg2-Atg18	[67]
	Atg9	[68,69,70,71]
	Atg18/Hsv2	[72,73]
**Quantitative Techniques**		
Isothermal Titration Calorimetry (ITC)	Atg18	[74]
	Atg21	[74]
Surface Plasmon Resonance (SPR)	Atg18	[31]
	Atg20, Atg24	[65]
**Techniques to Monitor Membrane Tethering and Restructuring**		
Dynamic Light Scattering (DLS) (Tethering)	Atg1	[52,53]
	Atg2	[12]
	Atg8	[51]
	Atg11	[62]
	Atg17	[53]
Turbidity Assay (Tethering)	Atg8	[51]
	Atg12-Atg5-Atg16	[54]
Bead Capture Assay (Tethering)	Atg1	[52]
	Atg2	[21]
Fluorescence Microscopy	Atg8	[51,54,63]
	Atg18	[56]
	Atg20-Atg24	[75]
	Atg23	[48]
Electron Microscopy	Atg8	[51]
	Atg9	[8]
	Atg11	[62]
	Atg17, Atg1 pentameric complex	[53]
	Atg20-Atg24	[57,58,59]
	Vac8	[34]
Lipid Mixing Assays	Atg2	[12]
	Atg8	[51]

## Data Availability

Not applicable.

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
