# Peer review of "Characterization of Protein–Membrane Interactions in Yeast Autophagy"

_cells, 2022, doi:10.3390/cells11121876_

Round 1

Reviewer 1 Report

The authors provide a comprehensive overview on the techniques used to analyze the membrane association of autophagy core proteins. I think this is in interesting summary especially useful for newcomers/students in the field. The review is clearly written and I only have a few comments:

Figure 2:

The Atg2-18 complex localizes to the rim of the isolation membrane and not to the back side.

The Atg1 initiation complex and PI3K complex should be at isolation membrane-vacuole contacts (PAS) including Vac8 at the back side of the isolation membrane.

The authors could also include the Atg1-Atg8 interaction of the isolation membrane.

The review could include Vac8 and its role in tethering Atg1 kinse complexes for the initiation of autophagosome biogenesis (Hollenstein et al., 2019 and 2021)

Table 1 should include Atg9 scramblase activity

Page 5 at the end of the second paragraph, the authors refer to fraction 13, which is oddly specific.

Page 10 first line: 300 ml of 100 um protein should say 100 mM

Page 10 second paragraph: “at increasing concentrations ranging from mM to nM” should probably say “nM to mM”

Author Response

We would like to thank the reviewers for their enthusiasm for the manuscript and their helpful suggestions. We have expanded the limitations of some techniques, revised Figure 2 and added a section about Vac8. Followed are our detailed responses (normal text) to the specific points raised by the reviewers (italics).

Reviewer #1

The authors provide a comprehensive overview on the techniques used to analyze the membrane association of autophagy core proteins. I think this is in interesting summary especially useful for newcomers/students in the field. The review is clearly written and I only have a few comments:

 Figure 2:

The Atg2-18 complex localizes to the rim of the isolation membrane and not to the back side.

The Atg1 initiation complex and PI3K complex should be at isolation membrane-vacuole contacts (PAS) including Vac8 at the back side of the isolation membrane.

The authors could also include the Atg1-Atg8 interaction of the isolation membrane.

We have updated Figure 2 to localize the Atg2-Atg18 complex to the rim of the isolation membrane. We have also added the vacuole and the interaction between the initiation complex and Vac8. We attempted to include more detailed features of autophagy including the interaction between Atg1 and Atg8 but we found that the figure became too convoluted. Therefore, we have produced a figure which represents the major features of autophagy. 

 The review could include Vac8 and its role in tethering Atg1 kinse complexes for the initiation of autophagosome biogenesis (Hollenstein et al., 2019 and 2021)

We thank the reviewer for this suggestion. We have added an additional section focusing on Vac8 and the suggested citations. We have included a new discussion on the roles of Vac8 in tethering the initiation complex to the vacuole and its later role in autophagosome-vacuole fusion.

Table 1 should include Atg9 scramblase activity

Page 5 at the end of the second paragraph, the authors refer to fraction 13, which is oddly specific.

Page 10 first line: 300 ml of 100 um protein should say 100 mM

Page 10 second paragraph: “at increasing concentrations ranging from mM to nM” should probably say “nM to mM”

All of these changes were made to the manuscript and tracked.

Reviewer 2 Report

Leary and Ragusa have put together a very thorough review on the particular topic of protein-membrane interactions in yeast autophagy. The authors first started with an introduction to the many techniques scientists have used to characterize these interactions and then, summarized the main results from the use of such approaches, alone or in combination, in the particular case of autophagy proteins. I find the review very nicely written, and timely relevant. I really enjoyed reading the review from the point of view of the membrane-protein interphase.

Major comment: I wonder why the authors did not include information about the events during autophagosome-vacuole fusion, where there are other protein-membrane interactions. I understand that the review may be already too long and with a focus on autophagy proteins, but if they find it relevant, maybe a comment could be welcomed by the readers interested in the whole pathway of autophagy, not only the early stages.

Minor comments: 1) Page 4: when discussing the pros and cons of fluorescence microscopy in the study of membrane-protein interactions, authors should contemplate including other limitations such as tags that may interfere with the localization, there are spatial resolution limitations, and a limit to the number of fluorophores to detect at the same time.

2) Page 5: First paragraph, please, format the references Mizushima et al and Wang et al (twice) properly.

3) Page 6: First paragraph, maybe mention that fixation during preparation of EM specimens can generate artifacts as a negative aspect of this technique.

4) Page 10: first line, change "100 um" to "100 uM", if the concentration of the protein is what you are referring to.

Author Response

We would like to thank the reviewers for their enthusiasm for the manuscript and their helpful suggestions. We have expanded the limitations of some techniques, revised Figure 2 and added a section about Vac8. Followed are our detailed responses (normal text) to the specific points raised by the reviewers (italics).

Reviewer #2

Leary and Ragusa have put together a very thorough review on the particular topic of protein-membrane interactions in yeast autophagy. The authors first started with an introduction to the many techniques scientists have used to characterize these interactions and then, summarized the main results from the use of such approaches, alone or in combination, in the particular case of autophagy proteins. I find the review very nicely written, and timely relevant. I really enjoyed reading the review from the point of view of the membrane-protein interphase.

Major comment: I wonder why the authors did not include information about the events during autophagosome-vacuole fusion, where there are other protein-membrane interactions. I understand that the review may be already too long and with a focus on autophagy proteins, but if they find it relevant, maybe a comment could be welcomed by the readers interested in the whole pathway of autophagy, not only the early stages.

We thank the reviewer for raising this point and agree a discussion of autophagosome-vacuole fusion would enhance the review. However, as the reviewer notes the article is already quite long and we feel that while it would be informative to add a more detailed discussion on autophagosome-vacuole fusion it would make the manuscript too long and potentially inaccessible. Instead, we have included a sentence emphasizing that our focus was on autophagosome biogenesis rather than autophagosome-vacuole fusion. Nevertheless, taking into consideration comments from both reviewers we have added a new section discussing Vac8, which is required for tethering the autophagy initiation complex and is also required for autophagosome-vacuole fusion.

Minor comments: 1) Page 4: when discussing the pros and cons of fluorescence microscopy in the study of membrane-protein interactions, authors should contemplate including other limitations such as tags that may interfere with the localization, there are spatial resolution limitations, and a limit to the number of fluorophores to detect at the same time.

2) Page 5: First paragraph, please, format the references Mizushima et al and Wang et al (twice) properly.

3) Page 6: First paragraph, maybe mention that fixation during preparation of EM specimens can generate artifacts as a negative aspect of this technique.

4) Page 10: first line, change "100 um" to "100 uM", if the concentration of the protein is what you are referring to.

We have included the suggested limitations to the fluorescence microscopy and EM sections and have made the additional changes pointed out by the reviewer.